# Recent Developments on the Catalytic and Biosensing Applications of Porous Nanomaterials

**DOI:** 10.3390/nano13152184

**Published:** 2023-07-26

**Authors:** Nabanita Pal, Debabrata Chakraborty, Eun-Bum Cho, Jeong Gil Seo

**Affiliations:** 1Department of Physics and Chemistry, Mahatma Gandhi Institute of Technology, Gandipet, Hyderabad 500075, India; nabanitapal_chem@mgit.ac.in; 2Institute for Applied Chemistry, Department of Fine Chemistry, Seoul National University of Science and Technology, Seoul 01811, Republic of Korea; debabratachem81@gmail.com; 3Department of Chemical Engineering, Hanyang University, Seoul 04763, Republic of Korea; 4Clean-Energy Research Institute, Hanyang University, Seoul 04763, Republic of Korea

**Keywords:** nanomaterials, porosity, stability, catalysis, biosensing

## Abstract

Nanoscopic materials have demonstrated a versatile role in almost every emerging field of research. Nanomaterials have come to be one of the most important fields of advanced research today due to its controllable particle size in the nanoscale range, capacity to adopt diverse forms and morphologies, high surface area, and involvement of transition and non-transition metals. With the introduction of porosity, nanomaterials have become a more promising candidate than their bulk counterparts in catalysis, biomedicine, drug delivery, and other areas. This review intends to compile a self-contained set of papers related to new synthesis methods and versatile applications of porous nanomaterials that can give a realistic picture of current state-of-the-art research, especially for catalysis and sensor area. Especially, we cover various surface functionalization strategies by improving accessibility and mass transfer limitation of catalytic applications for wide variety of materials, including organic and inorganic materials (metals/metal oxides) with covalent porous organic (COFs) and inorganic (silica/carbon) frameworks, constituting solid backgrounds on porous materials.

## 1. Introduction

The concept of nanotechnology was first introduced by the world-famous Noble Laureate Richard P. Feynman during his lecture “There’s Plenty of Room at the Bottom: An Invitation to Enter a New Field of Physics” at the annual meeting on 29 December 1959, when he tried to manipulate matter on an atomic scale [1]. Nanoparticles (NPs) and nanostructured materials, parts of the nanotechnology field, have recently been the focus of worldwide interest owing to their broad scopes of applications [2,3]. NPs are substances with external nanoscale dimensions in the 1–100 nm range and can be one (1D), two (2D), or three (3D) dimensional, depending on their shapes. Nanomaterials (NMs) are materials with an internal or external structure of nanoscale dimensions, and are generally composed of several agglomerated particles [4].

NMs and nanostructured materials have recently attracted interest for technological, biomedical, and socioeconomic applications, owing to their highly tunable physical and chemical properties [5,6]. Since the first discovery of MCM-41-type ordered mesoporous silica by scientists of Mobil corporation in 1992 [7,8], considerable progress has been made over 30 years in the synthesis of mesoporous materials with various structures and components [9,10]. Similar to hexagonal mesoporous MCM-41 and large-pore SBA-15 silicas, there has been a surge in the popularity of introducing porosity onto NMs matrices. NMs offer greater versatility in surface functionality, catalytic properties, and biosensing than their nonporous bulk counterparts [11,12]. Thus, widespread knowledge and financial investment have been directed towards the development of advanced and versatile applications for newly synthesized porous NMs [5,6,7,13,14]. Due to their innovative properties, porous nanomaterials in the form of silica- [15], carbon- [16,17,18,19], and noble/non-noble-metal-based [20,21,22,23,24] nanomaterials have been intelligently engineered for a wide range of uses, including catalysis [25,26,27,28,29,30], energy conversion and storage [31,32,33,34,35], biosensors [36,37,38,39,40,41,42,43,44,45], antibacterial [46], drug delivery [47], and gas uptake [48].

Consequently, many communications, research articles, reviews, and mini-reviews on the applications of porous NMs have been published in both national and international journals. These publications are readily available in the Web of Science database, indicating the high level of interest and activity in this field of research [1,2,3,4,5,6,7,8,9,10,11,12,13,14,15,16,17,18,19,20,21,22,23,24,25,26,27,28,29,30,31,32,33,34,35,36,37,38,39,40,41,42,43,44,45,46,47,48]. Recognizing the vast amount of research on porous NMs and the numerous applications that have been explored, we have chosen to narrow the focus of this review to the catalytic and sensing applications of these materials. In this review, we aim to provide an overview of the different types of porous NPs, including their syntheses and properties. Furthermore, we will examine the most important applications of these materials in the fields of catalysis and sensing.

The NMs can be classified according to their morphologies, dimensionalities, states, chemical compositions, and other factors. Depending on their morphologies, NMs can have a high or low aspect ratio (e.g., nanotubes and nanospheres). NMs can exist in dispersed, suspended, agglomerated, and other forms, based on their syntheses and surface functionalizations. NMs are broadly divided into four groups according to their dimensionality (0D, 1D, 2D, and 3D) [2,4]. To further illustrate the diversity of porous NPs, a flowchart has been provided in Figure 1 that outlines different categories of these materials. In addition to the porous NPs mentioned in the flowchart, it is also worth noting that nanomaterials can be classified based on their chemical composition into well-known classes, such as carbon-based nanomaterials, inorganic-based materials, organic-based nanomaterials, and composite-based nanomaterials. Some examples of carbon-based NMs, which are mainly constituted of carbon, are carbon nanotubes (CNTs) and graphene [49,50,51]. In contrast, inorganic NMs can have metallic (e.g., Ag, Cu, and Au) or metal oxide (e.g., TiO_2_, Al_2_O_3_, and ZnO) structures [4]. Inorganic-based NMs include mixed oxides such as ZnTiO_3_, NiAl_2_O_4_, and silica (SiO_2_). Furthermore, dendrimers, cyclodextrins, liposomes, and other materials formed from organic compounds should be considered as organic-based NMs. Composite NMs have multiple phases with at least one phase of nanoscale dimensions. Thus, combinations of porous metal oxides, silicas, oxides, and carbon can be considered composite-based NMs [52]. In this review, we aim to provide an overview (Figure 2) of the different types of porous NPs, including their syntheses and properties. Furthermore, we will examine the most important applications of these materials in the fields of catalysis and sensing.

## 2. Properties and Characterization of Porous NMs

Porous NMs find widespread applications in various fields such as daily commodities, industry, biomedicine, and chemical research [4,52]. The adequate application of NMs depends on the detailed determination of their physicochemical properties, including crystallinity, phase purity, morphology, particle size, surface properties, internal structure, thermal stability, reactivity, and biocompatibility. Several analytical techniques and instruments have been used to characterize NMs [53]. Some of the most common techniques and instruments used to determine the properties of NMs are listed in Table 1. In addition, specific analytical tools have been occasionally employed to study the application of NMs, such as gas chromatography–mass spectrometry for catalytic studies, UV-spectrometry for photocatalysis, cyclic voltammetry (CV) and amperometry for sensing analysis, physical property measurement systems to evaluate magnetic properties, and in vitro cell viability and in vivo microbial colony viability tests for biological studies.

## 3. Catalytic Applications of Porous NMs

NMs have a wide range of applications, owing to their unique properties. As covering all fields of application is unfeasible, this review highlights two of the most significant and extensively researched areas of NMs, namely, catalytic and biosensing applications. Porous NMs are versatile catalysts with controllable morphologies and local environments that allow for the optimization of catalytic activity [54]. Their tunable pore diameters and good surface areas make them suitable for catalytic beds. Moreover, these materials can be functionalized to alter the textural properties and polarity of the base matrix. Porous NMs possess favorable stability, bio-compatibility, and reusability towards heterogeneous catalysis for various advanced acid–base, redox, photocatalytic, organic, and environmentally benign reactions [11]. Magnetic porous materials have also been used as supports to prepare porous solid catalysts for several organic transformation reactions [55,56,57,58].

### 3.1. Silica and Silica-Supported Catalysts

In this section, we review recent reports on the catalytic properties of porous silica and silica-supported NMs.

#### 3.1.1. Functionalized Mesoporous Silica Nanocatalyst

Mesoporous silica NPs (MSNs) are widely used catalysts in the mesoporous family of materials (Figure 3) [47]. Porous silicas have extremely high BET surface areas and good pore volumes with tunable pore diameters, rendering them suitable for heterogeneous catalysis [59]. Although pure silica rarely exhibits appreciable catalytic activity, owing to its structural inertness, its catalytic properties can be improved by incorporating suitable metal ions or organic functional groups into the covalent Si-O framework. Rana et al. recently reported the synthesis of amine-functionalized silica using sonication. This material was an excellent catalyst for the Knoevenagel condensation of benzaldehyde and a malonic ester to produce cinnamic acid (Figure 4) [60]. A product selectivity of up to 95% obtained at room temperature was the highest among reported NH_2_-functionalized silicas. The catalyst could be reused three times without significant loss of activity [60]. The morphology of the silica NPs is crucial in the catalytic performance of functionalized silicas [61]. For example, silica hollow nanospheres (SHNs) with large core spaces can effectively accommodate the reagents required for catalytic reactions, and their porous shells help them diffuse into the core.

In 2013, Li et al. reviewed the role of organo-functionalized SHNs in chiral catalysis and cascade reactions [62]. The 4-(dialkylamino)pyridine-functionalized mesoporous silica nanospheres synthesized by Chen et al. proved to be efficient nucleophilic catalysts for Baylis–Hillman, acylation, and silylation reactions, presenting high reactivity, reusability, and selectivity [63].

MSNs immobilized with both acidic and basic groups have been successfully used as bifunctional heterogeneous catalysts for C-C bond formation [64]. Shylesh et al. reported the one-step synthesis of amine- and sulfonic acid-functionalized silica nanospheres and investigated their cooperative role in various organic coupling reactions, such as nitroaldol and deacetalization–aldol reactions [64]. Furthermore, bifunctionalized MSNs successfully catalyzed the synthesis of HMF from bio-based carbohydrates, such as glucose, fructose, sucrose, and starch-based biomolecules, enabling the environmentally friendly conversion of biomass to an alternative energy source [65].

#### 3.1.2. Porous Silica Supported Metal-Doped Catalysts

The energy crisis, or the depletion of fossil fuels, is one of the world’s most important issues. Furthermore, the combustion of fossil fuels contributes significantly to environmental pollution by releasing CO_2_ into the atmosphere. As a result, hydrogen gas production via water splitting via electricity has been regarded as the most pure and abundant energy source in this decade [66,67,68]. Several electrocatalysts have been produced by researchers worldwide, but there are several limitations such as high cost, low durability, and high toxicity [69,70,71]. Porous silica materials have no such disadvantage and also demonstrate outstanding electrocatalytic activity towards the over water splitting reaction, which is currently the greatest method for clean energy production [72,73,74,75,76,77,78]. Very recently, Meng et al. have created an in-situ CoP-doped Co_3_(Si_2_O_5_)_2_(OH)_4_ nanosheets material for bifunctional electrocatalysis, i.e., hydrogen evolution reaction (HER) and oxygen evolution reaction (OER). This electrocatalyst (CoP/CSNSs) displayed an overpotential of 251 mV@10 mA/cm^2^ current density with respect to a reference hydrogen electrode (RHE) in a 1 M KOH electrolyte for HER, and, surprisingly, the activity of the silica composite was equal to a commercially available Pt/C electrocatalyst at a high current density (∼150 mA/cm^2^) [79]. Metal nanoparticle-, metal oxide- or mixed-oxide-based silica NMs can be synthesized by co-condensation or post-impregnation of the silica framework with suitable reagents. Metal-based silica NMs can be versatile catalysts, depending on the metal ions incorporated into the silica matrix. For example, Ce(IV)O_2_-loaded silica nanostructures oxidized benzyl alcohol to benzaldehyde at room temperature under solvent-free conditions [80]. Furthermore, Ce(IV)O_2_/Ce_2_(III)O_3_-loaded silica mesoporous composites exhibited both redox and acidic properties for the oxidation of hydrocarbons and acid-catalyzed acylation of aromatic and aliphatic alcohols, respectively [81]. Periodic mesoporous organosilicas (PMO) grafted with different transition metals, such as Ti, V, Cr, and Mo, have been used in various chemical reactions [82,83,84]. In 2022, Chatterjee et al. prepared a novel PMO using a Schiff base precursor synthesized by the condensation of p-terphenyl-4,4′-dialdehyde and APTMS [85]. The PMO was grafted with Ag NPs to form Ag@PMO, which has been shown to be an excellent catalyst for the oxidation of styrene to styrene oxide using CO_2_ as an oxidant under mild reaction conditions (Figure 5) [85]. Transition metal oxide NPs supported on silica materials also exhibit versatile catalytic performance towards different organic transformations. Kankala et al. reported the encapsulation of metal species into MSNs and their application in catalysis [86]. Magnetic-nanoparticle-doped silica (Fe_3_O_4_@Silica) is an ideal support material for the immobilization of Pd and Ag NPs in order to create suitable catalysts for the reduction of organic molecules, allowing for simple catalyst separation using a strong magnet [87].

Morphology-oriented silica nanocatalysts have been extensively applied in heterogeneous catalysis [61]. The catalytic activity and product yield significantly depended on the structure of the NMs. Diacon et al. recently incorporated iron oxide into silica hollow spheres and studied their catalytic performances in the Fischer–Tropsch conversion of CO to CO_2_ [88]. According to the Polshettiwar group, fibrous silica nanospheres have distinguishable catalytic activities when loaded with different metal ions, such as Pd, Ru, and Pt, owing to their exceptionally unique microstructures. Therefore, they are excellent candidate catalysts for various organic reactions [89]. Metal-oxide-loaded silica core–shell nanostructures have also proven to be efficient catalysts for hydrogenation and other catalytic reduction reactions [90,91]. The use of porous materials to prepare the immobilized enzymes has been extensively investigated in the past years, and this material showed excellent catalytic activities towards several organic transformation reactions [92,93]. In 2016, Xie et al. used a surfactant-directed sol-gel technique to create a Fe_3_O_4_-MCM-41 nanocomposite capable of immobilizing lipase on a core-shell structure. After immobilizing the lipase, the heterojunction nanocomposite has been utilized as a magnetically separable biocatalyst for the interesterification reaction in-between soybean oil and lard [94].

#### 3.1.3. Porous Phyllosilicate Nanocatalysts

Supported metal (oxide) NPs have gained interest in the last decade because of their energy and/or selectivity in oxidation and (de)hydrogenation reactions, which are the most common catalytic reactions used in pollution control, energy, and chemical industries [95]. However, their lower thermal stability limits their catalytic potential. In 2013, Dumitriu et al. developed a novel method for the synthesis of metal (oxide) NPs in a phyllosilicate (PS) structure with improved dispersion and stability, which are useful for catalytic applications [96]. PS, also known as sheet silicate, is a mineral class that includes mica, chlorite, serpentine, and talc and is characterized by continuous tetrahedral and octahedral sheets. The tetrahedral sheets have a central cation, T, coordinated to four tetrahedral oxygen atoms, three of which are linked to the adjacent tetrahedra, forming a boundless broadened hexagonal sheet along the 2D crystallographic direction (Figure 6a). In contrast, in octahedral sheets, the cation is situated at the center, coordinating with six anions (e.g., F, Cl, O, and OH) and sharing six corners with the neighboring octahedron (Figure 6b). A common plane (Figure 6c) was formed by the tetrahedra, with the free corners pointing to the same side of the sheet [97].

Kaolinite is composed of one tetrahedral sheet and one octahedral sheet (1:1), and Smectite is composed of two tetrahedral sheets sandwiched between one octahedral sheet (2:1). The Si^4+^, Al^3+^, and Fe^3+^ central cations (T) lead to the formation of tetrahedral sheets, whereas Mg^2+^, Mn^2+^, Ni^2+^, Co^2+^, Cu^2+^, and Zn^2+^ form octahedral sheets. As shown in Figure 6, the sheets can bend and roll because of the similar molecular sizes of silicon tetrahedrons and metal octahedrons. Several PS-based materials have recently been applied in metal-ion batteries [98], supercapacitors [99], and metal-ion adsorbents. The chemical, environmental, and energy industries primarily use PSs as catalysts or catalyst precursors. The catalytic properties of Ni, Cu, Co, and Ce PSs were recently examined. Among metal silicates, Ni-based silica materials have been widely studied because of their high abundance, activity, cost-effectiveness, and availability of silica supports with varying pore structures. However, there are some limitations in the synthesis of these catalysts, such as the need for high temperatures (>180 °C), long reaction times (>24 h), and excess nickel reagents due to the lack of surface silanol groups. Thus, Ni utilization was lower than those of the silica materials. Liu et al. developed a double accelerator method to synthesize Ni-PS materials at lower temperatures (40 °C) without using an excess Ni precursor [100]. Furthermore, Cho et al. recently reported the first mesoporous spherical Ni-PS (Ni/Si = 1) with excellent structural integrity after 8 weeks of hydrothermal (100 °C) treatment (Figure 7) [101]. Ni-PSs are commonly used for the catalytic conversion of methane, a greenhouse gas, to syngas through partial oxidation or dry reforming processes because of their superior stability and carbon resistance. These reactions correspond to steam and dry (CO_2_) reforming (SRM and DRM) and partial oxidation of methane (POM) methods. Sivaiah et al. reported the use of pure Ni-PS in a DRM process with excellent catalytic conversion at 700 °C for 12 h [102]. The presence of a larger number of hydroxyl (-OH) groups on the PS surface suppressed carbon formation. A different Ni-PS supported on porous silica nano-spheres was reported by Yang et al. [103] for the POM at 700 °C for 50 h. TEM imaging of the material after the catalytic oxidation process revealed a homogeneous distribution of Ni-NPs over the specimen without carbon deposition. The water–gas shift reaction (WGS) is commonly combined with methane steam reforming in downstream processes to balance the H_2_/CO ratio. This is one of the most important reactions in the chemical industry, allowing the production of methanol, ammonia, and hydrocarbons, along with H_2_ gas. 

The three main reactions involved in methane gas reforming are as follows:(1)H_2_O + CH_4_ → CO + 3H_2_(2)CO_2_ + CH_4_ → 2CO + 2H_2_(3)O_2_ + 2CH_4_ → 2CO + 4H_2_

The Fe-Cr catalyst, known as a high-temperature shift catalyst (HTS), was previously used as a catalyst for this reaction at moderately high temperatures (583–683 K). However, the high toxicity of chromium led to the development of alternative Ni catalysts. Despite their high catalytic performance, Ni-based materials have seldom been applied in industry for HTS because of their greater tendency towards methanation (Figure 8), which suppresses the production of H_2_ [104]. In Figure 8, Kawi et al. showed the effect of doping of alkali metal (Na) in the Ni-based catalysis towards methane suppression during HTS reaction. The pyrolysis of PSs at different temperatures produced a Ni/SiO_2_ composite material with extraordinary catalytic activity towards HTS [105]. Bi-metallic (Ni-Mg) PS nanotubes were also reported for the high-temperature (650 °C) WGS reaction [106]. Cho et al. also reported the HDO reaction of m-cresol in atmospheric hydrogen pressure catalyzed by Ni/silica and Nickel silicate (Ni-MCM-41) [107]. Sintering of the Ni particles was observed with the traditional Ni/SiO_2_ catalysts at high Ni loadings. To circumvent this, Liu et al. synthesized a series of Ni-PS materials via hydrothermal treatment of mesoporous silica nanorods and a Ni(II) salt and impregnation of the CeO_2_ promoter into the framework. This CeO_2_-modified Ni-PS exhibited excellent catalytic activity for the methanation of CO and CO_2_ above 350 and 600 °C, respectively, for 6 h [108]. Cu-PS has recently been developed for the hydrogenation of dimethyl oxalate (DMO) because of its superior stability, catalytic activity, and selectivity [109]. On the other hand, Gong et al. developed a Cu/SiO_2_ catalyst that hydrogenated DMO to ethanol using Cu PS as a catalyst [110]. CO_2_ improved the activation achieved by Cu, making it an excellent hydrogenation catalyst for the production of methanol from CO_2_. Wang et al. also developed a Cu-PS porous silica material for the same catalytic process [111]. Syngas (CO+H_2_) can be converted into sulfur-free liquid hydrocarbon fuels via the Fischer–Tropsch reaction, a process that uses Co-based catalysts.

However, the lower dispersion with increasing Cobalt percentage (20%) limits its application. In 2017, Park et al. synthesized a Co/SiO_2_ catalyst from a Co-PS framework for high-temperature Fischer–Tropsch reactions [112].

### 3.2. Metal-Oxide- and Phosphate-Based Catalysts

Although porous oxide and phosphate nanostructures have smaller surface areas than silica, they provide a great platform for heterogeneous catalytic transformations because of their high crystallinity, thermal stability, and number of active sites [113,114,115,116,117,118,119]. Akbari et al. reviewed the applications of metal-oxide NPs from groups 4 to 9 and 11 for the selective oxidation of alkenes, alcohols, and aldehydes [120].

Ren et al. reported highly crystalline 3D mesoporous oxides such as Cr_2_O_3_, Co_3_O_4_, Fe_2_O_3_, CuO, and β-MnO_2_ for catalytic CO oxidation reactions [118]. Self-assembled TiO_2_ NPs also proved to be highly efficient heterogeneous catalyst supports for various organic reactions. De et al. reported a straightforward sol–gel synthesis of a well-defined spherical TiO_2_ nanocatalyst using aspartic acid and its use in the dehydration of D-fructose and D-glucose to HMF under microwave heating [121]. The high density of acid sites in the TiO_2_ nanospheres and high NP surface area led to a high catalytic yield of HMF.

Sarkar et al. recently prepared boat-, dumbbell-, and cuboid-shaped cerium hydroxidophosphate materials via sol–gel-mediated hydrothermal synthesis and employed them for the oxidative coupling of thiols to disulfides in the presence of H_2_O_2_ and air [113]. Bifunctional Zn-Ti-based nanocatalysts exhibited catalytic activities in oxidation and benzylation reactions, demonstrating the high efficiency and stability of this heterogeneous support for organic transformations [122]. Pramanik et al. have demonstrated the crucial role of organic–inorganic iron–phosphate NPs in transesterification reactions for biofuel synthesis under mild reaction conditions [123]. The reusable catalyst was prepared from benzene-1,3,5-triphosphonic acid and FeCl_3_ via hydrothermal synthesis. Mixed-oxide NPs are superior heterogeneous catalysts to single oxide NPs because of their higher densities of catalytic acidic or basic sites [124]. In 2010, Paul et al. employed newly designed mesoporous nickel aluminate mixed-oxide NPs for the liquid-phase catalytic reduction of an aromatic nitro compound [125]. Compared to single-oxide alumina (0.6% yield), the binary Ni-Al oxide system exhibited improved conversion efficiency (approximately 50% yield) under similar reaction conditions (Figure 9). A similar phenomenon was observed for a Mg-Al mixed oxide, where the presence of Lewis basic sites catalyzed the formation of lactones and esters from cyclic and acyclic ketones via Baeyer–Villiger oxidation [126].

Abdelrahman et al. reported various stoichiometric oxides with excellent efficiency and reusability for the photocatalytic degradation of organic dyes using UV irradiation [127].

### 3.3. Pure Organic and Organic–Inorganic Hybrid Nanocatalysts

Pure organic NMs solely composed of covalent bonds (C-C, C-N, C-O, and B-O) are known as covalent organic frameworks (COF) or porous organic polymers (POP). These materials exhibit distinct synthesis and structural properties [128,129,130]. Generally, COFs are (semi-) crystalline NMs produced through reversible condensation reactions. In contrast, POPs are amorphous materials with a clear-cut porous architecture produced by irreversible condensation reactions such as C-C coupling [131]. In the year 2023, Banerjee et al. reported three metal-free porous imine-based COFs with remarkable stability (Figure 10) in the year 2023, enabling C-H borylation at ambient temperature and nitrogen pressure under blue-light-emitting diodes [132]. Different functional groups, e.g., -CH_3_, -Ph, -Cl, -OH, are compatible with this catalytic process. They demonstrated photocatalytic activity as a function of several physical and photophysical properties, such as BET surface areas, optical property, and band gaps of three separate COFs with different building blocks employed.

They have shown 12 different substrate scopes, including quinolines, pyridines, and pyrimidines with moderate yields (up to 96%). Among all three COFs, TpAzo-COF shows the best catalytic performance due to its high surface area and low bandgap compared to the others. The most important analytical tool for investigating the photo-activity of a material is charge separation efficiency, which is also investigated thoroughly in this article by performing fluorescence and electrochemical techniques. This finding points global scientists in a new approach toward sustainable and environmentally friendly societal growth.

Chowdhury et al. recently reported an N-rich porous organic polymer (POP) synthesized from three different olefin monomers, with excellent catalytic activities for the synthesis of cyclic carbonates from epoxides, using CO_2_ as the carbon source [133].

Porous coordination polymers, metal–organic frameworks (MOFs), PMOs, and metal phosphonate homoleptic open frameworks are emerging hybrid organic–inorganic porous NMs. Among these, MOFs have been explored for heterogeneous organocatalytic applications, owing to the presence of “molecular scaffolds” in the periodic architecture [134,135,136,137,138].

MOFs are constructed from metal-ion nodes and organic compound bridging linkers. Over 90,000 MOFs have been reported in the literature, with 500,000 characterized structures [139,140,141]. This includes different organic ligands such as thiolates, carboxylates, phosphonates, imidazolates, and oxalates. There are three main active sites in the MOFs responsible for catalytic activity: open metal sites, defect metal sites, and organic linkers [142]. A Cu_3_BTC (BTC = 1,3,5 benzene tricarboxylic acid) framework was synthesized by Nikseresht et al. and used to produce tacrine analogs using ultrasonication. The study revealed that the Cu metal sites play a crucial role in catalysis instead of the ligand, providing an example of the open metal site as an active catalytic site [143].

In 2020, Park et al. reported a CO_2_ fixation reaction using a Cu(II)-based MOF as the catalyst. The role of Cu was investigated using theoretical studies [144]. An example of an MOF containing defect metal active sites was reported by Caratelli et al. in 2017 using two different hydrated- and dehydrated-linker-deficient UIO-66 and UIO-66-NH_2_. This MOF was applied in Fischer esterification reactions using carboxylic acids and methanol as substrates. The intermediate of this catalytic reaction is stabilized by H-bonding with extra Brønsted acid sites in the hydrated framework [145]. An overview of the different functionalities integrated into MOFs and the role of the linker in catalytic selectivity (for Baylis–Hillman, click, acetal, alcohol oxidation, and other reactions) was recently demonstrated by Cao et al. represented in Figure 11 [146].

In conjunction with metal binding and covalent postfunctionalization, the authors displayed an all-round multifaceted MOF synthesizing procedure for creating multivariate heterogeneous organocatalysts. They have successfully integrated secondary and ternary ligands into the pristine LIFM-28 and then dynamically disintegrated also. Two or more catalytic sites have also been accurately and quantitatively formed into the MOF co-ordination spheres for sequential reaction. These findings suggest that MOFs are suitable as multifunctional catalysts, which are potential techniques to enhance the efficiency and environmental friendliness of heterogeneous catalysis. Chakraborty et al. reported a Ni-W mixed metal phosphonate open framework material for the photoelectrochemical oxygen evolution reaction (PEC-OER) in 1 M KOH, achieving an O_2_ evolution rate of 275 μmol·g^−1^ [147]. The same group developed a tetradentate phosphonate ligand-based Co-MOF for the electrochemical hydrogen evolution reaction in different solvents, including seawater. DFT-VASP studies were performed to establish the structure–catalytic property relationship during hydrogen evolution. H_2_ production rates of 4.5, 2.3, 1.8, and 1.5 μmol·g^−1^ were obtained in 0.5 M H_2_SO_4_, pH = 4, pH = 6, and seawater media, respectively [148].

### 3.4. Composite-Nanomaterials-Based Catalyst

Composite nanoporous materials possess properties that are distinct from those of their individual components [52]. Thus, oxide–oxide and oxide–silica composites are commonly used, industrially important, and ecofriendly heterogeneous catalysts. Zhang et al. prepared various morphology-oriented porous composites of CuO/Cu_2_O oxides using lauric acid as the capping agent [149]. The catalytic oxidation of CO to CO_2_ over these composites significantly depended on their morphologies, and a significant improvement was observed when the morphology changed from cubic to octahedral and from rod-like to wire. The oxidation of CO over porous oxide composites has also been reported by another group of scientists [150]. Well-dispersed hollow microspheres of CeO_2_-ZnO oxide composites were synthesized by Xie et al. and loaded with Au NPs for the catalytic oxidation of CO at lower temperatures [151].

In 2012, Xu et al. synthesized an ordered porous NiO-CaO-Al_2_O_3_ composite using sol-gel-mediated EISA, which showed excellent catalytic performance in the CO_2_ reforming of methane gas [152]. Mesoporous oxide–silica composites, also known as silica/ceria–silica composites, are active catalysts for heterogeneous liquid-phase catalytic reactions and have been applied in the solvent-free oxidation of benzyl alcohol to benzaldehyde at room temperature with 50% conversion (Figure 12) [80].

MOF-based composite materials have lately gained popularity due to their high porosity, which allows for the production of host–guest composite materials. Unlike the MOF-based catalysts described above, which use metal MOFs as catalysts or catalyst components that remain intact throughout the reaction time, MOFs can serve as self-sacrificing templates for fabricating highly porous carbon-based materials by pyrolyzing at different temperatures (most commonly between 700 and 1000 °C), and these materials have been explored for a variety of catalysis applications [153,154,155,156]. In 2018, Huo et al. reported the efficient and site-selective oxidation of diols and hydrogenates using Pt/ZIF-8 by restricting the physical space within MOF pores [157]. Furthermore, Kennedy et al. developed a Ru-immobilized Zr-based MOF, which is a highly active and stable CO_2_ methanation catalyst [158].

### 3.5. Porous Carbon-Based Nanocatalyst

Porous carbon-based NMs are superior catalyst beds for heterogeneous catalysis because of their high surface areas, large pore volumes, and controllable pore sizes [159,160]. Porous carbon nanostructures loaded with various metal NPs can catalyze multiple industrially important organic transformations. Recently, Zhang et al. reported a novel Ni-embedded porous carbon catalyst, synthesized via hydrothermal synthesis, for the catalytic cracking of biomass tar [161]. Moreover, Gogoi et al. described the ability of Cu and Co nanoparticle-decorated porous carbon materials to reduce nitroaromatics to aniline derivatives using NaBH4 [160]. The high efficiency of the catalyst, as well as its reusability, eco-friendliness, easy separation using a strong magnet, and the ability to provide an environmentally friendly reaction pathway were also demonstrated. De et al. previously reviewed the synthesis of porous carbon NMs from biomass and explored their applications in emerging catalytic reactions [161].

### 3.6. Porous Metal-Based Catalyst

In recent years, nanoporous metals have been considered as one of the most unique members in the nanoporous family due to their high BET surface area, uncommon porosity, and excellent electrical conductivity. These properties also make them extremely promising candidate for a broad range of important applications (e.g., energy storage, sensing, and catalysis) [162,163,164]. This material is highly efficient for electrocatalyst for the oxidation of small molecules, e.g., methanol, ethanol, and formic acid, and it is also used for oxygen reduction reaction. All of those reactions play critical roles in fuel cell applications [165]. Porous platinum metal electrode has been used mostly for this purpose [165]. To decrease the cost of this Pt-based electrode, researchers used transition metals (e.g., Fe, Cu, Ni, and Co) containing porous bimetallic electrode materials. More notably, bimetallic Pt-based catalysts are appealing for the following reasons: (1) Due to the synergistic actions of both components, bimetallic catalysts can considerably improve resistance to carbon monoxide poisoning for the oxidation of tiny organic molecules. (2) Due to electrical, alloying, or strain effects, the addition of a second metal can increase catalytic activity [163]. An electrospinning process combined with chemical alloying was used by Shui et al. to create nanoporous Pt-Fe bimetallic alloy nanowires with overall wire diameters of 10–20 nm and a ligament diameter of 2–3 nm [166]. Due to the presence of chemically active metal surfaces on those, core–shell-like composite materials have also been prepared by using alloyed nanoporous metals for better catalytic applications. For example, Li and Ding demonstrated that the one-step oxidation of np-Ag by hydrogen peroxide (H_2_O_2_) within the addition of HCl could yield AgCl-coated np-Ag (AgCl/np-Ag) composite catalysts, showing excellent photocatalytic degradation activity towards the methyl orange dye [167]. The most interesting thing is that this material could show plasmonic properties, and this property was helpful to absorb the light from the UV to infrared regions. That is why this material is considered as highly capable for photocatalysis [168].

## 4. Applications of Porous NMs in Biosensing

The biosensing or sensing of biological agents and toxic chemicals in physiological systems is an important part of biomedicine [169,170,171]. NMs have recently been explored in the biosensing field because of their unique physicochemical properties, including a high surface area, tunable pore diameter, and variable oxidation state of metal-based NPs [172,173]. NPs mainly sense biomolecules through colorimetric, fluorescence, and electrochemical sensing.

### 4.1. Colorimetric Biosensing

The colorimetric detection of biomolecules, such as H_2_O_2_, glucose, and antioxidants, is vital for the diagnosis of diseases and analysis of food safety. Porous metal-based and metal NPs supported on porous silica or carbon matrices can act as artificial nanozymes (NMs that mimic enzymes). In 2011, Kim et al. reported the synthesis of a porous silica-based nanocomposite embedded with Fe_3_O_4_ magnetic NPs that mimicked peroxidase activity [174]. When loaded with any oxidative enzyme, the composite generated H_2_O_2_ from the target molecule. The generated H_2_O_2_ reacts with Fe_3_O_4_ to form a colored compound that could be detected colorimetrically (Figure 13). A similar principle was applied to the detection of other target molecules by changing the enzyme used. Wang et al. recently investigated the detection of glutathione, H_2_O_2_, glucose, and other components using Fe NP incorporated into 2D carbon nanosheets [175]. The Fe@CNs behaved like a dual enzymatic system, mimicking the re-activity of oxidase and peroxidase to calorimetrically detect the substrate 3,3′,5,5′-tetramethylbenzidin. The colorimetric sensing of dopamine in beef can also be achieved using CuS-modified bovine-serum-albumin-functionalized copper phosphate NPs [176]. Au and Pt NPs can also be used as nanozymes for the colorimetric sensing of biomolecules [177,178].

### 4.2. Porous NMs in Fluorescence Biosensing

The advanced optoelectrical properties of NPs have prompted their widespread applications as fluorescent biosensors for detecting wastewater pollutants [179,180]. In this process, NMs are conjugated with biomolecules, such as enzymes and antibodies, for the selective detection of target species. Gaviria-Arroyave et al. previously reviewed the fluorescence biosensing of environmental pollutants [179]. Furthermore, Yaraki et al. recently examined the utilization of metal NPs on supported porous matrices as improved fluorescence biosensors [173]. NPs have also been widely used to fabricate FRET-based biosensors to monitor and detect the behaviors of DNA and RNA in biological systems [181]. The chemical immobilization of porous silica is straightforward because of the presence of surface -OH functional groups. Varma et al. employed this concept to develop a sol–gel nanoporous silica substrate with a high surface area for biosensing cardiac markers [182]. Chakraborty et al. developed a tetratopic phosphonate linker (1,1,2,2-Tetrakis(4-phosphonophenyl)ethylene)-based Mn-MOF single crystal for the remediation of sepsis by selectively sensing arginine over lysine and other biofluids in aqueous media [183]. Simple, rapid, and highly sensitive DNA detection methods are required for early clinical diagnostics and screening of genetic disorders. Through non-covalent (П-П) interactions between POP nanospheres and DNA, Liu et al. developed a simple and efficient fluorescent biosensing platform capable of detecting multiplex DNA [184]. A covalent organic framework (COF) is a crystalline organic porous architecture comprising reversible condensations of building materials that have a highly ordered structure and regulated porosity [185]. COFs have been employed for biosensing applications since 2014 [186,187,188]. Pharmaceutical enterprises, hospitals, nursing homes, and families are the primary sources of antibiotic discharges into water bodies as a result of COVID-19 and the current pandemic. Detecting and removing them from bodies of water is vital, but it can be difficult. Zhong et al. created triazine-based COF nanosheets that functioned as a fluorescence-induced biosensor to detect nitrofurans (effective antibiotics). There was a LOD of 4.97 ppb for nitrofurazone, 8.08 ppb for nitrofurantoin, and 13.35 ppb for furazolidone [189]. An immunoassay based on fluorescent sensors was developed by Liu et al. to detect various malignancies with a GOx@ZIF-8 composite signal-transduction tag in 2017 [190].

### 4.3. Electrochemical Biosensing Using Porous NMs

Transition metals and metal oxides exhibit a wide range of catalytic properties, owing to their multiple possible oxidation states. Porous metal oxides, metal composites, and metal–silica mixed oxides are thus extensively used for the electrochemical sensing of biomolecules such as glucose, H_2_O_2_, uric acid, ascorbic acid, and dopamine [191].

Electrochemical biosensors transduce biochemical information into electrical signals via current, voltage, or impedance modulation. The electrodes, which are the main components of these analytical devices, are modified with NMs, whereas the solid support is functionalized with biomolecules. The performance and sensitivity of biosensors depend on their nature, functionalization, surface area, and biocompatibility, as well as the immobilization of the biomolecules [192]. Nanostructured materials employed in biosensor preparation can be divided into two groups: carbon-based and non-carbon-based (silica and other metal oxides) biosensors.

### 4.4. Carbon-Based NMs for Biosensors

Porous silicon electrochemical biosensors offer practical, simple, low-power, cost-efficient, and low-maintenance alternatives to optical biosensors. However, despite their potential, they remain underdeveloped and underused. Several novel approaches for the functionalization of porous NMs are currently under development, including the derivatization, oxidation, and preparation of porous NMs nanocomposites. These nanocomposites exhibit high BET surface area, tunable porous architectures, and high biocompatibility. The Hong group employed modified gold electrodes based on nanoporous silicon as novel urea detection biosensors [193]. Glucose sensing is vital for diabetes diagnosis. Despite their high sensitivities and selectivities, natural enzymes are not ideal for glucose detection, owing to their high cost, poor long-term stability, and difficult immobilization processes. Alternative glucose sensors that combine light and acoustic wave technologies with electrochemical sensors are currently being developed. Simple, low-cost, highly sensitive, and rapid electrochemical glucose sensors are highly appealing. In 2019, Yamauchi et al. described the application of hierarchical Ni-BDC nanosheets for electrochemical glucose sensing (Figure 14) at very low concentrations (LOD = 6.68 μm) [194]. Porous carbon NPs are widely used as solid supports for enzymatic and non-enzymatic biosensors owing to their high surface area. Single-walled CNTs, multi-walled CNTs, graphene, and graphene oxide have been used in biosensors [195]. In 2021, Gupta et al. employed CNT-based microelectrodes for the enzyme-free biosensing of glucose with very high sensitivity. The loading of copper NPs into CNTs improved glucose electrooxidation, leading to CV and amperometry responses at very low LODs [196]. Kang et al. reported the importance of enzyme glucose oxidase (GOD) and the high biocompatibility of chitosan in GOD–graphene–chitosan-modified electrodes for glucose detection [197]. The electrochemical detection of H_2_O_2_ was also successfully performed by Shin et al. who modified reduced graphene oxide (rGO) with Au NPs and horseradish-peroxidase-encapsulated protein NPs to prepare highly sensitive electrodes for the selective detection of H_2_O_2_ in the presence of interfering agents such as glucose and uric acid [198]. The albumin protein used in this study can retain a significant amount of HRP. Furthermore, rGO significantly improved electron transfer, and the Au NPs increased the surface area and electrical properties. The HEPNP/rGO/Au working electrodes exhibited high sensitivity, high selectivity, and a low LOD for H_2_O_2_ biosensing in human blood serum. The detection of uric acid in urine and other biological fluids has important practical healthcare implications. Carbon nanofibers synthesized using phosphate lignin can be used as wearable and flexible biosensing platforms to selectively identify uric acid in artificial urine. Baig et al. recently reported the fabrication of graphene-based electrodes as disposable sensing platforms to detect uric acid with high selectivity and sensitivity [199].

Covalent organic framework (COF) topologies and unique designs will be influenced by the next generation of electrochemical sensors and biosensors [200,201,202]. In 2019, Du et al. synthesized a novel COF (TBAPy-MA-COF-COOH) via a polycondensation method using 1,3,6,8-tetra(4-carboxylphenyl)pyrene and melamine as substrates. After several surface modifications, they introduced Cu_2_O@AuNPs and AgNCs@AuNP into the COF’s architecture, and this has been used as an electrochemical biomarker for miRNA 155 and miRNA 122 [203]. The COF material was easily exfoliated into 2D nanosheets with 2–4 nm thickness values. Using these features, Wang et al. reported a bimetallic-incorporated, COF-based composite nanosheet for electrochemical detection of levodopa, which is used to treat Parkinson’s disease [204]. Severe acute respiratory syndrome coronavirus 2 (SARS-CoV-2) remains a leading cause of severe health problems worldwide. In 2021, Zhang et al. developed a bifunctional electrochemical biosensor that could detect the SARS-CoV-2 N-gene with high sensitivity using porphyrin-based POPs [205]. Recently, MOF/COF are widely used for biosensing through enzyme immobilization than the others because they show excellent biocompatibility, tunability, and well-known crystallinity [206]. The high surface areas of those materials are also helpful for the loading of the enzyme in a different weight ratio. The enzyme immobilization procedure has become easier for those materials, as they have structural and functional varieties. To best of our knowledge there are some review paper on this topic [207,208,209]. An electrochemical biosensor platform based on GDH and ZIFs was attempted by Mao et al. to monitor glucose levels. In this study, they showed the sensing limit 0.1 to 2.0 mM, which is very high due to the low conductivity and less affinity of MOFs [210]. In later studies, many methods have been proposed by researchers to increase affinities as well as electronic conduction [211]. High sensitivities and low background noise have made the photoelectrochemistry biosensors attractive [212]. Yb-MOFs created by Li and colleagues with ionic liquids with large conjugate systems and coordinations with Yb demonstrated a strong near-infrared PEC response. When Au NPs were reduced on the Yb-MOF surface, incoming light was absorbed by Yb-MOF, and electron–hole pairs were separated more quickly. Upon recognition of its target by the CEA antibody on the surface of Yb-MOF@Au-NPs, CEA is adsorbed after the photocurrent density is gradually decreased due to restrictions of electron–hole pair separations in the composite materials [213]. As a result of poor science or technology, bacterial contamination has become one of the most important issues in many nations [214]. In 2017, Ranjbar et al. developed a novel electrochemical biosensor using EDC-NHS chemistry and aptamers immobilized in modified ZIFs-8, and, after that, it was modified by ferrocene–graphene oxide heterojunction through П-П interaction. The final composite materials have been used as an electroactive indicator for the detection of *Pseudomonas aeruginosa* (*P. aeruginosa*) bacteria [215]. The electrochemical characterization was monitored using cyclic voltammetry and electrochemical impedance spectroscopy methods. In this study, the authors used differential plus voltammetry techniques to detect the corresponding bacteria (Figure 15) with a low detection limit of 1.2 × 10^1^–1.2 × 10^7^ CFU mL^−1^.

### 4.5. Non Carbon-Based NMs for Biosensors

Various porous metal, metal-oxide, and metal-doped silica and oxide–silica nanocomposites have been successfully employed as electrochemical biosensing platforms for the detection of biomolecules [191,216]. For example, nanostructured composites of self-assembled NiTiO_3_/NiO particles have been used as sensitive enzyme-free platforms for the electrochemical detection of glucose. The high surface area of the nanocomposites and the enhanced redox properties of the metal ions facilitated the immobilization and electrooxidation of the glucose analyte, leading to a low LOD in the presence of interfering agents [217]. Similarly, a metal-incorporated silica (Cu-SBA-15)-modified electrode with 5% Cu loading (Si/Cu = 20) exhibited a good and selective response for glucose in CV and amperometric analysis, with a linear behavior in the 10–20 µM range and an LOD of 10 µM [218]. The high sensitivity of the material was attributed to the synergistic effect between the metal ion and the high surface area porous silica, which ensured an optimal platform for metal ion-analyte interaction. A Ni-doped silica with nickel hydrosilicate (Ni_3_Si_2_O_5_(OH)_4_), presenting a yolk-shell morphology, also exhibited excellent performance for the non-enzymatic electrochemical detection of glucose at room temperature (Figure 16) [219].

Metal oxides and hydroxides are also good electrochemical glucose sensors, as reported by Bhaumik et al. Although both Ni(OH)_2_ and NiO nanostructures have been used for enzyme-free glucose sensing, NiO exhibits a comparatively higher sensitivity than its hydroxide analog [191]. Lu et al. developed porous ZnO-nanosheet-based microspheres and employed them in an electrochemical H_2_O_2_ biosensor with linearity in 1–410 and 10–2700 μM ranges [220]. Furthermore, few reviews have highlighted the importance of graphene oxide, silica, and others in biosensing [221,222,223]. Nanoporous gold (NPG) could be a practical contender as an enzyme-free biosensor for electrochemical location with great affectability and selectivity since gold has excellent catalytic activity towards oxidation or reduction of some tiny organic molecules, e.g., glucose, H_2_O_2_, and others. The deployed NPG thin films showed outstanding electrical current activity for glucose oxidation [224]. In this, they have also shown the effect of pore size on non-enzymatic glucose sensing very nicely. The materials with the smallest pore size (18 nm) achieved the greatest enhancement. An NPG-electrode-based electrochemical DNA biosensor was developed by Lin et al. for the identification of the promyelocytic leukemia/retinoic acid receptor α (PML/RARα) fusion gene in acute promyelocytic leukemia (APL), where methylene blue was used as electroactive indicator. Here, they applied differential plus voltammetry techniques for sensing, achieved a very low detection limit, e.g., 6.7 pM [225]. Nanoporous non-noble metals are more alluring as an electrode material for chemical sensor applications in terms of cost-effectiveness. Nanoporous cupper (NPC), having pore sizes in between 100 and 200 nm, has been synthesized from the Al_60_Cu_40_ alloy by using a conventional dealloying process in the presence of 5 wt% HCl. With the help of an adsorption technique, Horseradish peroxidase (HRP) has been immobilized into the porous architecture of that Cu metal. Due to its high electric conductivity, along with its activity it has been deployed as an electrochemical biosensor for O-phenylenediamine (OPD) with 0.37 μA μM^−1^ sensitivity [226].

## 5. Summary and Future Prospect

Nanoporous materials (NMs) can be divided into zeolites, mesoporous materials, metal–organic frameworks (MOFs), covalent organic frameworks (COFs), and porous organic polymers (POPs). In this review, we present versatile applications using nanoporous materials, especially for catalysts and biosensors. In addition, there are so many commercial ones for industrial applications. Especially, MOFs have been growing very rapidly for various applications due to their high surface areas, up to 3000 m^2^/g, and large number of species using versatile combination in the synthesis routes. However, except for a few types of zeolites (e.g., MFI type ZSM-5), it is still difficult to use the other types of nanoporous materials as catalysts through mass production because they do not have structural stability in wet conditions at high temperatures to be used as catalysts [227,228,229]. Mesoporous materials were invented in an effort to convert hydrocarbon compounds into other useful compounds through chemical treatment in oil refineries and were predicted to be more efficient catalysts than zeolites in treating large hydrocarbon compounds. Therefore, since 1992, many studies have been conducted for about 30 years to secure various reactions and efficiencies as catalysts for mesoporous materials, but a material whose practicality has been verified for structural stability that guarantees economic feasibility has not yet been secured. The hydrophobic periodic mesoporous organosilica (PMO) structure may be a suitable material for a catalytic reaction at a low temperature of about 100–200 °C; however, stability at high-pressures and high-temperatures over 400 °C is not guaranteed because of the low thermal stability of the organic components. This review article provided an overview of the catalytic applications of porous nanomaterials as well as the impact of porosity and the presence of metal species on their catalytic performance. Furthermore, it provides information on the role of metals, composites, silica, and carbon-based nanostructures in biosensing applications, including the effects of the specific properties of the transition metals and NM morphologies on their catalytic and sensing properties. In particular, recently introduced mesoporous phyllosilicate structures have been confirmed to have higher structural stabilities than other crystalline mesoporous materials, and their efficacies as catalysts have also been demonstrated in several other papers [98,99,100,101,102,103,104,105,106,107,108]. In this review paper, we would like to draw attention to researchers in this field by introducing recent papers related to the synthesis and application of mesoporous phyllosilicate structures. It is thought that a new crystalline structure such as phyllosilicate can be an alternative that can be a breakthrough in practical aspects, even if the mesoporous material does not have a uniform pore structure. In addition, MOF structures are materials that have already secured efficiency and practicality, and, among the structures currently being studied, relatively interesting materials are collected and presented in this review.

Finally, this work serves as a guide for those interested in studying the catalytic and sensing properties of newly designed porous NMs.

## Figures and Tables

**Figure 1 nanomaterials-13-02184-f001:**
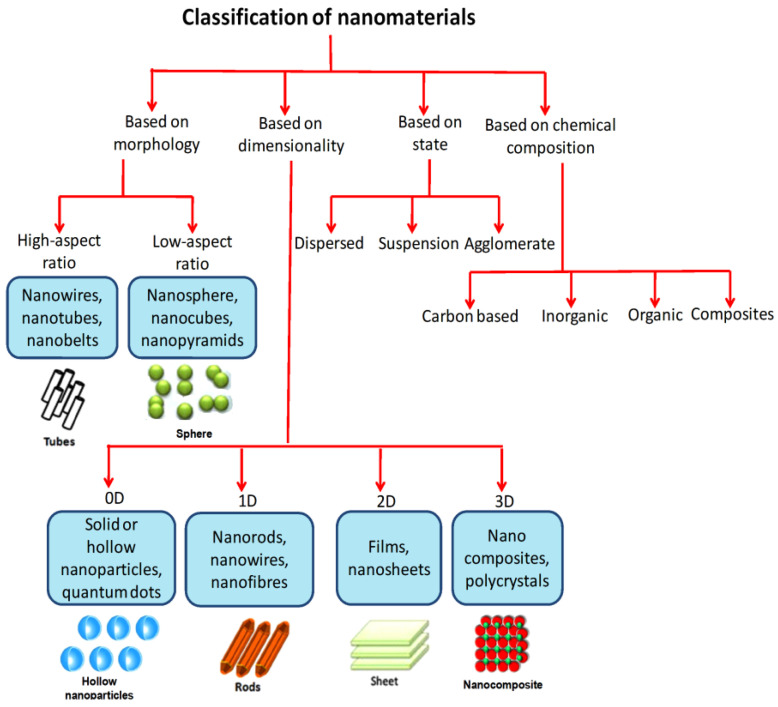
Classification of the NMs based on different categories.

**Figure 2 nanomaterials-13-02184-f002:**
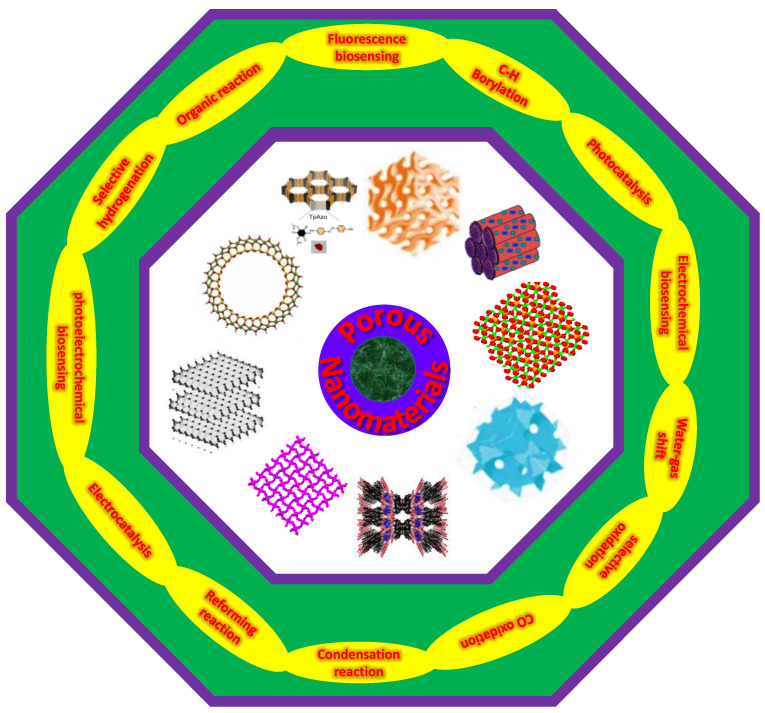
Pictorial representation of the corresponding research background illustrated in this review article.

**Figure 3 nanomaterials-13-02184-f003:**
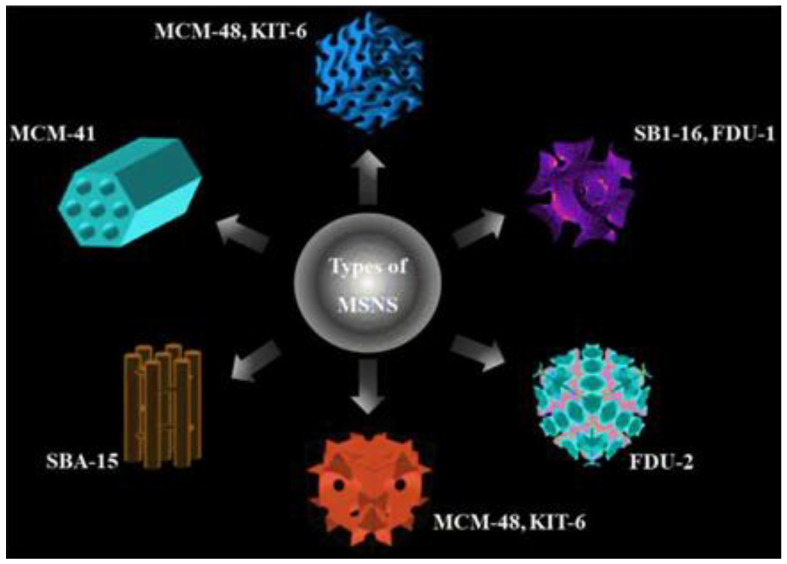
Illustration of different types of MSNs. Reprinted with permission from Narayan et al. [47].

**Figure 4 nanomaterials-13-02184-f004:**
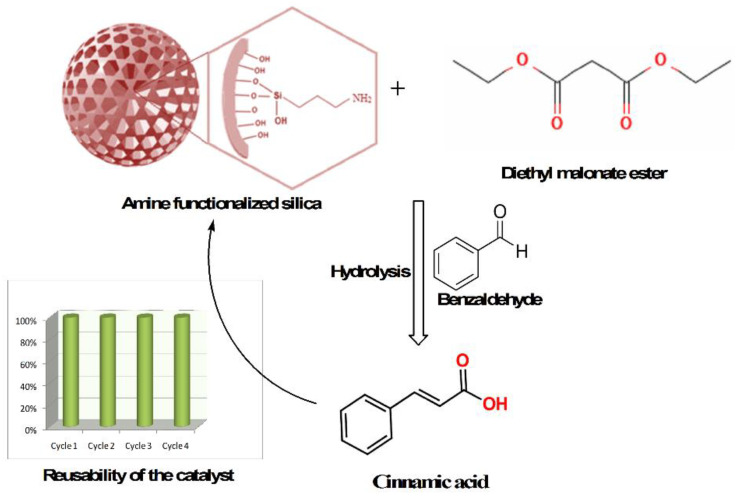
Knoevenagel condensation of benzaldehyde with diethyl malonate catalyzed by NH_2_-functionalized porous silica NMs [60].

**Figure 5 nanomaterials-13-02184-f005:**
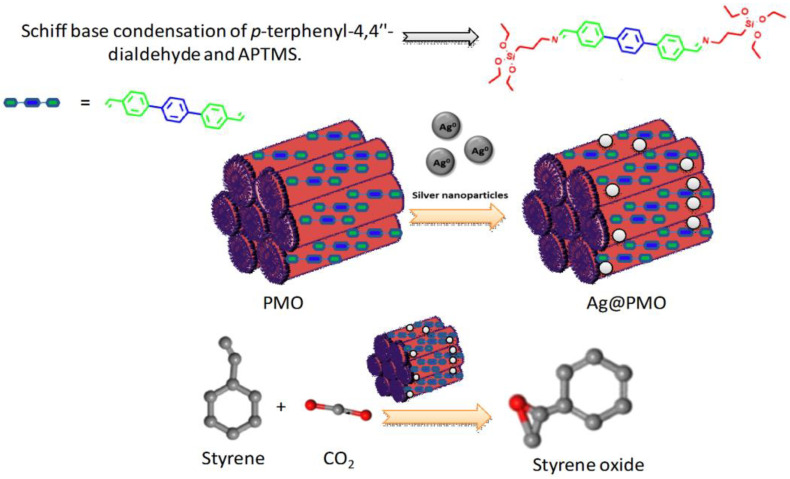
Synthesis of Ag@PMO and its application in the oxidation of styrene [85].

**Figure 6 nanomaterials-13-02184-f006:**
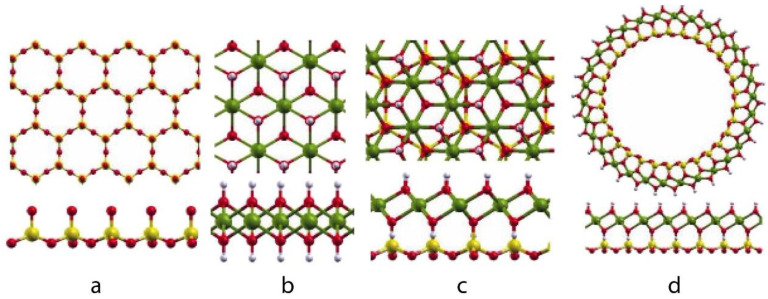
Top and side views of (**a**) SiO_2_, (**b**) Mg(OH)_2_, and (**c**) Mg_3_Si_2_O_5_(OH)_4_ layers and (**d**) a Mg_3_Si_2_O_5_(OH)_4_ nanotube. Atom labels: Si, yellow; O, red; H, white; Mg, green. Reprinted with permission from Duarte et al. [97].

**Figure 7 nanomaterials-13-02184-f007:**
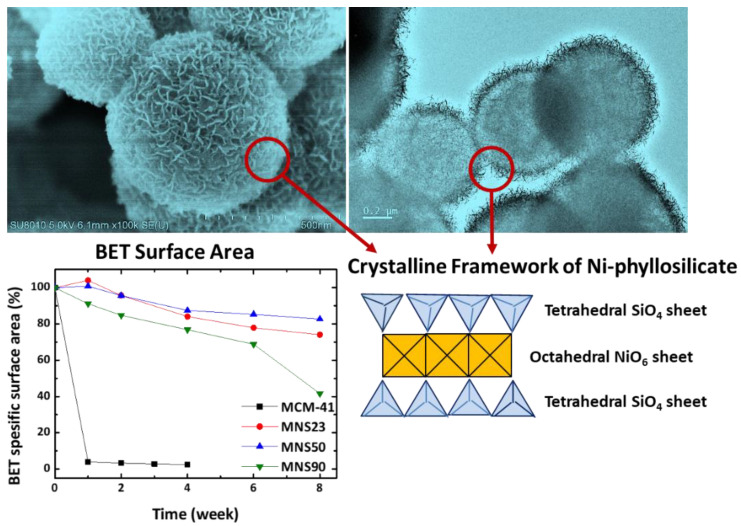
Hydrothermal stability of mesoporous nickel-PS particles. Reprinted with permission from Cho et al. [101].

**Figure 8 nanomaterials-13-02184-f008:**
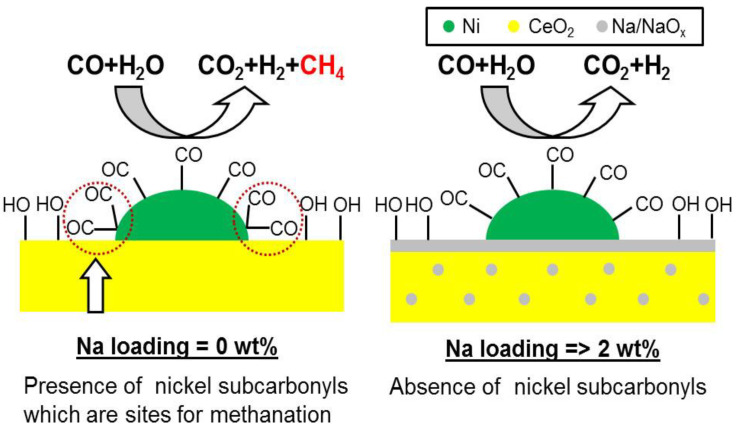
Methane suppression in Na-doped Ni/CeO_2_catalysts is depicted schematically. Reprinted with permission from Kawi et al. [104].

**Figure 9 nanomaterials-13-02184-f009:**
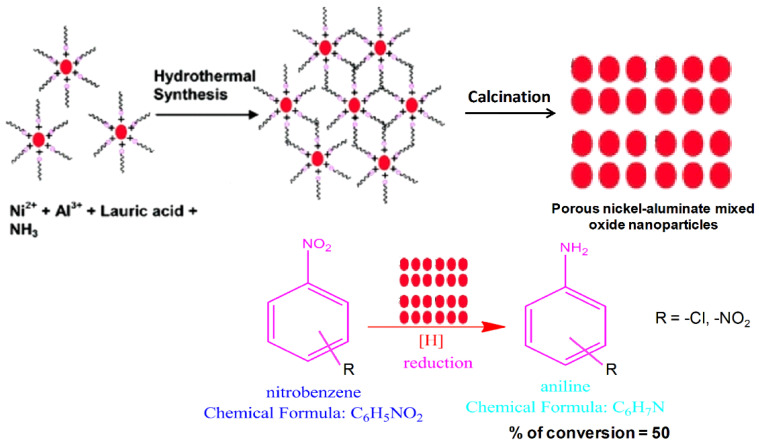
Reduction of nitrobenzene derivatives catalyzed by a porous Ni-Al mixed oxide [125].

**Figure 10 nanomaterials-13-02184-f010:**
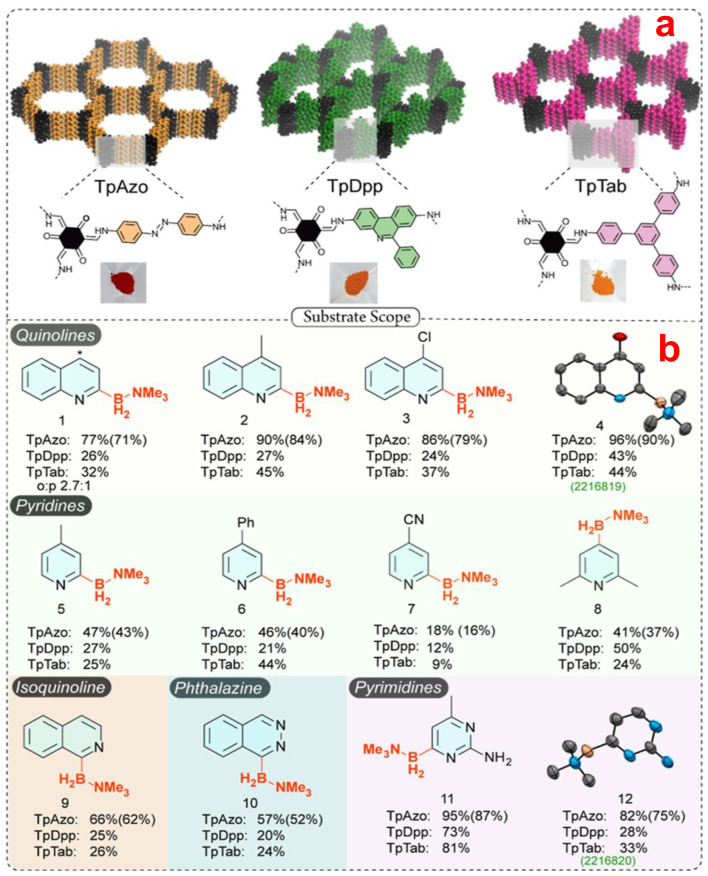
(**a**) Three different COF material structures have been synthesized by varying the amine linker. (**b**) Substrate scope for the photocatalytic metal-free C-H borylation reaction by utilizing the aforementioned COF structure (2 mg) under blue LEDs. Reprinted with permission from Banerjee et al. [132]. Borylation of quinoline (1) produced a mix of C_2_ and C_4_ (*) products, with a clear preference for the more active C_2_ site (C_2_/C_4_ = 2.7:1).

**Figure 11 nanomaterials-13-02184-f011:**
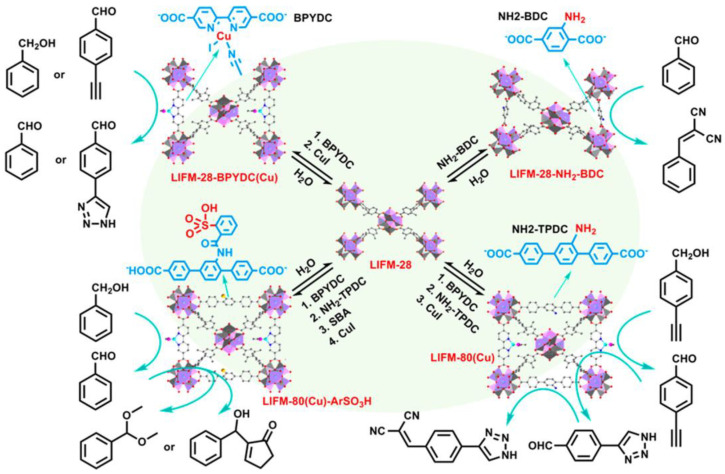
Multivariate MOF catalysts produced from proto-LIFM-28 using the dynamic spacer installation (DSI) method demonstrating their interconversion for various catalytic applications. Reprinted with permission from Su et al. [146].

**Figure 12 nanomaterials-13-02184-f012:**
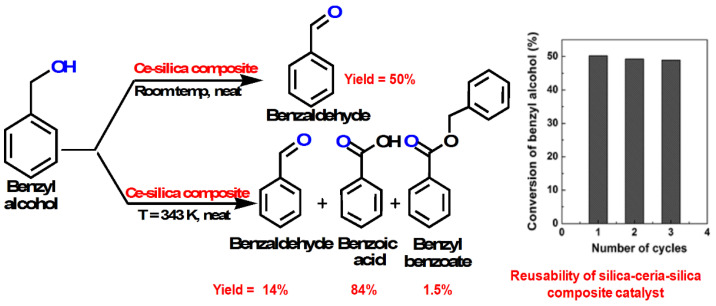
Catalytic activity of a porous silica/ceria–silica composite for the oxidation of benzyl alcohol [80].

**Figure 13 nanomaterials-13-02184-f013:**
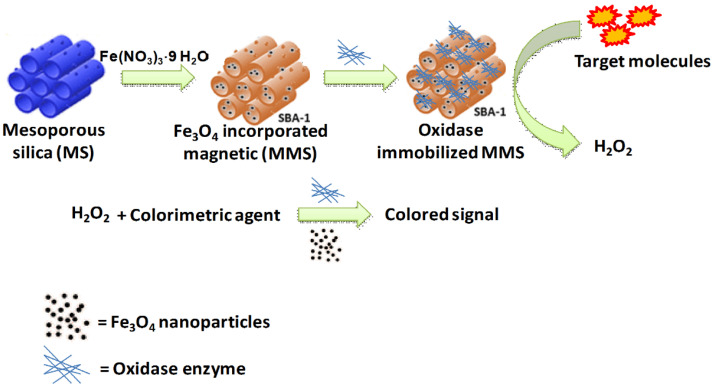
Fe_3_O_4_-loaded mesoporous silica for the colorimetric biosensing of target molecules in the presence of peroxidase [174].

**Figure 14 nanomaterials-13-02184-f014:**
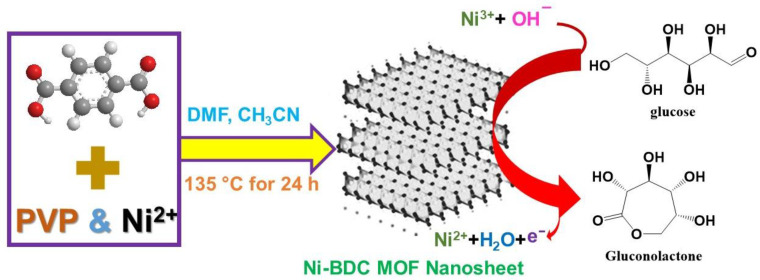
Electrochemical glucose sensing mechanism of Ni-BDC hierarchical sheet-like sensors [194].

**Figure 15 nanomaterials-13-02184-f015:**
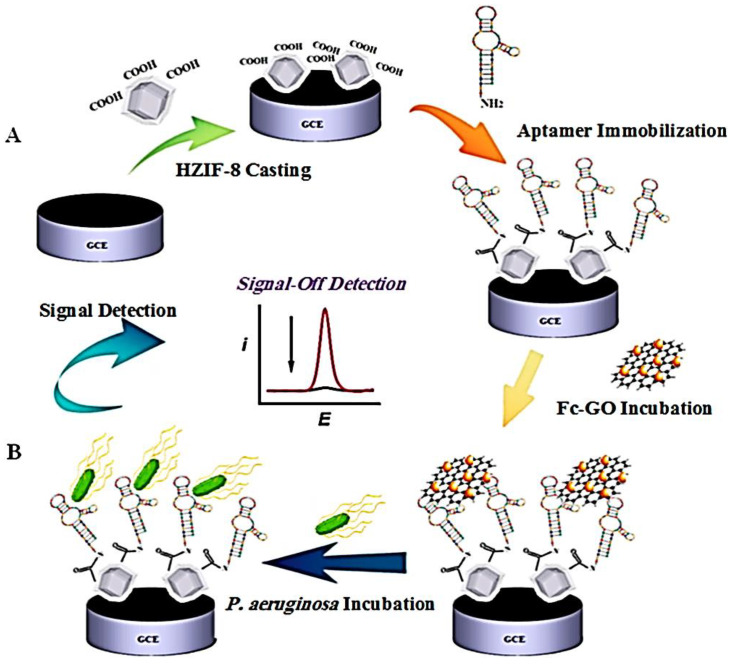
Schematic representation of ZIF-8 based composite fabrication (**A**) and detection of *Pseudomonas aeruginosa* bacteria [215] (**B**).

**Figure 16 nanomaterials-13-02184-f016:**
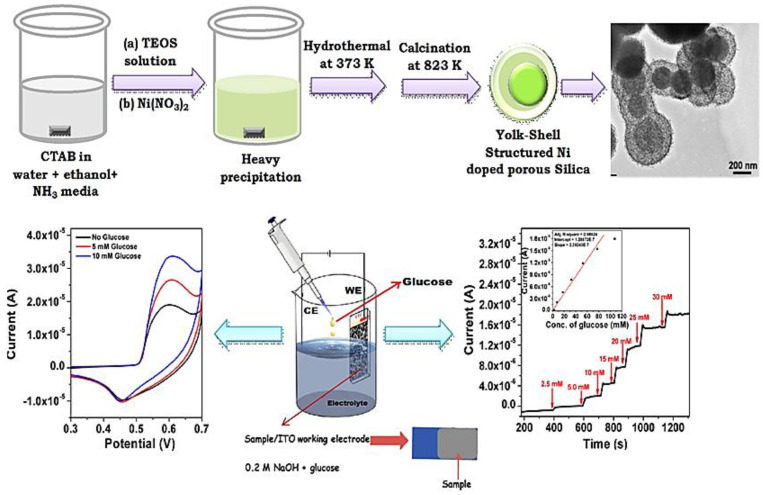
Synthesis of yolk-shell Ni-silica nanostructures and its electrochemical glucose sensing activity.

**Table 1 nanomaterials-13-02184-t001:** Analytical techniques and instruments used for the characterization of porous NMs.

Analytical Technique	Instrument	Properties
Powder X-ray diffraction (small and wide angle)	Powder X-ray diffractometer	Mesostructure, porosity, phase purity, and crystallinity
Brunauer–Emmett–Teller (BET) surface area analysis	BET surface area analyzer	Surface area, porosity, pore-diameter, pore volume, and shapes of pores
Transmission electron microscopy (TEM)	Transmission electron microscope	Internal nanostructure, particle size, pore crystallinity, and aggregation
Scanning electron microscopy (SEM)	Scanning electron microscope	Morphology, particle size and distribution, shape, and aggregation
Atomic force microscopy	Atomic force microscope	Particle size and distribution, shape, structure, and aggregation
X-ray photoelectron spectroscopy	X-ray photoelectron spectroscope	Oxidation state and chemical composition of surface
Fourier transform infrared spectroscopy	Fourier transform infrared spectroscope	Chemical bonding and bonding connectivity
UV-visible spectroscopy	UV-visible spectrophotometer	Chemical environment
Thermogravimetric analysis	Thermogravimetric analyzer	Thermal stability
Dynamic light scattering	Dynamic light scattering instruments	Size distribution

## Data Availability

Data sharing not applicable.

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
