# Peer review of "Recent Developments on the Catalytic and Biosensing Applications of Porous Nanomaterials"

_nanomaterials, 2023, doi:10.3390/nano13152184_

Round 1

Reviewer 1 Report

I studied this review manuscript with great interests. The authors carefully outlined the synthesis and structure of the nanoporous and nanostructured materials as well as their applications in heterogeneous catalysis and biosensing. This manuscript could be considered for publication after addressing the following questions.

(1) The classic references related to the first reports of representative materials such as MCM-41, SBA-15, COF, etc.

(2) Some figures provide abundant information but could not be explained in the text, for instance, Fig. 10. It would be great if more detailed discussion could be reorganized.

(3) Several figures possess poor resolution, e.g., Fig. 11, which can be improved in the revision stage.

(4) Furthermore, part of figures cannot be cited with corresponding reference (e.g., Fig. 15).

Reviewer 2 Report

In this review paper, interesting knowledge has been reported and it could be a contribution to the review on existing knowledge in the field. However, some revisions are required before it could be considered for publication as follows:

1. section 1 and 2 can be merged into one section, also so do as for section 3 and 4. On the contrary, section 5 is recommended to be divided to different sub-sections.

2. The literature review on the magnetic porous materials as supports to prepare porous solid catalysts, needs to be substantially enhanced referring to the published literature, such as 10.1016/j.rser.2022.113017; 10.1016/j.renene.2020.12.129; 10.1016/j.fuel.2020.118893; 10.1016/j.renene.2018.03.010.

3. In section 5, the use of porous materials to prepare the immobilized enzymes, has been extensively investigated in the past years. This aspect should be reviewed in this paper. Please referring the recently published articles for example: 10.1016/j.foodchem.2017.01.082; 10.1016/j.foodchem.2015.09.009; 10.1016/j.enconman.2018.01.021.

In this review paper, interesting knowledge has been reported and it could be a contribution to the review on existing knowledge in the field. However, some revisions are required before it could be considered for publication as follows:

1. section 1 and 2 can be merged into one section, also so do as for section 3 and 4. On the contrary, section 5 is recommended to be divided to different sub-sections.

2. The literature review on the magnetic porous materials as supports to prepare porous solid catalysts, needs to be substantially enhanced referring to the published literature, such as 10.1016/j.rser.2022.113017; 10.1016/j.renene.2020.12.129; 10.1016/j.fuel.2020.118893; 10.1016/j.renene.2018.03.010.

3. In section 5, the use of porous materials to prepare the immobilized enzymes, has been extensively investigated in the past years. This aspect should be reviewed in this paper. Please referring the recently published articles for example: 10.1016/j.foodchem.2017.01.082; 10.1016/j.foodchem.2015.09.009; 10.1016/j.enconman.2018.01.021.

Reviewer 3 Report

In this review the authors claim that they  intend to compile a self-contained set of papers related to new synthesis methods and versatile applications of porous nanomaterials that can give a realistic picture of current state-of-the-art research, especially for catalysis and sensor area. Especially, they would like to cover various surface functionalization strategies by improving accessibility and mass transfer limitation of catalytic applications for a variety of materials including organic and inorganic materials (metals/metal oxides) with porous organic (COFs) and inorganic (silica/carbon) frameworks constituting solid background on porous materials.

the idea to review this particular topic could be good, even if many other papers reported on the same topic. Anyway, the manuscript is really badly organized and several sections sound fully out-of-topic.

The title is "Recent Developments on the Catalytic and Biosensing Applications of Porous Nanomaterials" and the reader expects to find an overview on the porous materials and only on them. On the contrary there is a large part of the manuscript on nanoparticles and their composite (for example to me the section 3.1 is completely out of scope and it is really not clear why the authors mention for example the ref. 48 on NiO-ZrO2 and not many other REAL porous materials that can be fabricated)

The same for section 3.2. the manuscript should be about porous nanomaterials and not about nanoparticles (on the topic of nanoparticles there are hundreds of reviews and papers). For example, also in this section I don't see the rationale behind the choice to cite ref. 62.

Then the sections of applications of porous NMs. This part is a little bit more in line with the topic of the paper, but it sounds more like a review on Silica based materials, while silica is just of case of a material that can be prepared with porous structure. The authors also claim (line 169) that porous NM possess favorable stability, low toxicite and air-insensitivity, but this depends on the material! several porous material could react to oxidation and could be not stable!

the section of catalysis is rather confused. Also here it seems that the paper is more on silica based materials and not on general porous material. the section is rather long, but to me it is really not clear if they are discussing porous nanomaterials or composite materials mainly based on nanoparticles. 

the final section on sensing is rather short , while there is a huge literature on this specific topic.

Several figures have very low quality and must be improved

In the last section, the authors claim "Nanoporous materials can be divided into zeolites, mesoporous materials, metal-organic frameworks (MOFs), Covalent organic framework (COFs) and porous organic polymers (POPs). Except for a few types of zeolites, it is difficult to use them as catalysts  through mass production because they do not have structural stability to be used as catalysts."

this is rather wrong! nanoporous materials can be prepared using a large set of methods and materials. in particular the authors are here completely missing to mention nanoporous metals which find important applications in catalysis and sensing! moreover, being this a paper on porous materials it is very important to avoid to discuss on something out of the scope!

I think that the manuscript, in order to be considered for publication, needs a radical revision, in particular the authors must extend the discussion on porous materials including the important family of nanoporous metals.

some important references on the topic are:

Journal of Materials Science volume 54pages949–973 (2019)
Nature Reviews Chemistry volume 3pages108–118 (2019)
Chem. Soc. Rev., 2019, 48, 2366-2421
ACS Nano 2021, 15, 4, 6038–6060
Chem. Commun., 2022,58, 747-770
Chem. Soc. Rev., 2022,51, 2031-2080

and many others

english language is ok

Reviewer 4 Report

This paper reviewed the catalytic and biosensing applications of porous nanomaterials, which has potential application value in engineering. In order to meet the requirements of high-quality publication of the journal (Nonomaterials), it is recommended to consider the following suggestions.

1) Even though it's a Review, there are some key data recommended in the abstract.

2) The background section of the abstract is too long and needs to be simplified. 

3) The lenght of the introduction section is too short, and the innovation points of this review are not reflected.

4) It is suggested to add a figure describing the research background of this paper in Section I.

5) The quality of the red arrows in Figure 1 needs to be improved.

6) Is it appropriate to use color for the colors in Table 1?

7) Section 5 cannot just be a pile of literature, it needs to have its own analysis, especially its own comparative figures and tables.

8) It is best to have a discussion section.

9) There are too many outlooks that need to be refined.

10) There are few references in the last three years.

Round 2

Reviewer 3 Report

the authors have improved significantly the manuscript according to the referees' comments. 

I can now recommend the publication

some grammar errors are still in the text, but can be corrected during the proof correction step

Reviewer 4 Report

The authors have addressed all my concerns.